# FRICATIVE PHONEME DETECTION WITH ZERO DELAY

## ABSTRACT

People with high-frequency hearing loss rely on hearing aids that employ frequency lowering algorithms. These algorithms shift some of the sounds from the high frequency band to the lower frequency band where the sounds become more perceptible for the people with the condition. Fricative phonemes have an important part of their content concentrated in high frequency bands. It is important that the frequency lowering algorithm is activated exactly for the duration of a fricative phoneme, and kept off at all other times. Therefore, timely (with zero delay) and accurate fricative phoneme detection is a key problem for high quality hearing aids. In this paper we present a deep learning based fricative phoneme detection algorithm that has zero detection delay and achieves state-of-the-art fricative phoneme detection accuracy on the TIMIT Speech Corpus. All reported results are reproducible and come with easy to use code that could serve as a baseline for future research.

## 1 FRICATIVE PHONEMES AND HIGH-FREQUENCY HEARING LOSS

High-frequency hearing loss is a particular type of hearing loss that is mostly age related. People with high-frequency hearing loss have difficulties with word recognition in conversations. This is often caused by the loss of discrimination or detection ability of their ear for high-frequency sounds. Fricative phonemes are particularly hard to recognize for the affected people because these phonemes contain high-frequency information that is crucial to be able to discriminate them. The problem can persist if the affected people wear modern hearing aids. Researchers suggested frequency lowering approaches to make fricative phonemes audible by the hearing impaired. These techniques shift the high frequency parts of these sounds into lower frequency bands. For example, Robinson et al. (2009) show that a frequency transposition technique leads to improved detection and recognition ability in high frequency hearing impaired people for the sounds /s/ and /z/, which are fricative sounds. In this paper, we use the standard phonetic codes when we refer to the fricative phonemes as listed in Table 1. In order to employ such techniques in hearing aids, one must be able to reliably detect fricative phonemes with zero delay. This means that the algorithm needs to be able to detect that a fricative phoneme is coming next, *before* this has happened, so that the necessary alterations to the sound can be made in *real time*. In this paper we show how to use deep learning to solve this problem.

Table 1: Fricative phonemes of English.

| **PHONETIC CODE** | /s/ | /sh/ | /f/ | /th/ | /z/ | /zh/ | /v/ | /dh/ |
|---|---|---|---|---|---|---|---|---|
| **EXAMPLE WORD** | sea | she | fin | thin | zone | azure | van | then |

## 2 DATASET, NETWORK ARCHITECTURE, AND TRAINING

We employ a fully convolutional neural network with residual connections (He et al., 2016). The network takes raw speech segments as inputs, and calculates the posterior probability for the event that the sample at the end of the segment belongs to a fricative phoneme. The detection procedure is explained in Figure 1 where the raw speech segment is in blue, the input segment of the network is highlighted in green and the dashed green lines show the borders of the phonemes as marked in the TIMIT dataset (Garofolo et al., 1993). Based on the input segment, the network detects the identity

(belongs to a fricative phoneme or not) of the sound sample represented with the red vertical line at the end of the segment.

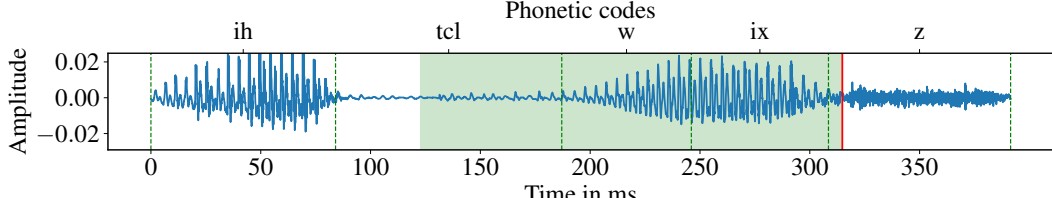

Figure 1: Speech segment pronouncing "it was" with the network input $x$ highlighted in green.

It is important to stress that the red line is at the end of the green segment, and not in the middle of the green segment. This means that we are not using non-causal information to detect the identity of the red sample. In other words, our method introduces zero statistical delay.

Observe that in Figure 1 the green segment spans four phonemes: /tcl/, /w/, /ix/, /z/ (see Table A.1 for an explanation of these phonetic codes). The red sample belongs to the phoneme /z/, which is a fricative phoneme, so that the correct detection in this case is *fricative*. Note that the fricative phoneme /z/ is barely contained in the green segment. Essentially, the network must learn to make accurate inferences about future phonemes based on the currently observed phonemes. Statistically, this is harder than making a detection about, for example, the middle point of the segment: think about the difference between interpolation and extrapolation.

## 2.1 DATASET AND DATA PREPROCESSING

The TIMIT Speech Corpus consists of raw speech sentences (at 16 kHz sampling rate) and their corresponding time aligned phonetic labels (Garofolo et al., 1993) as illustrated in Figure 1. It contains a total of 6300 sentences, 10 sentences spoken by each of 630 speakers from 8 major dialect regions of the United States. We used the entire training set of the TIMIT dataset for training (462 speakers), and the core test set for testing (24 speakers). We created the validation set (50 speakers) following the methodology in Halberstadt (1998).

In the TIMIT dataset, there are two sentences that are read by all the speakers; these are called *SA sentences*. We excluded these two sentences from all of the speakers in all sets (training, validation and test) to prevent a possible bias in the evaluation. Specifically, if SA sentences are not excluded, the model might conceivably memorize the locations of the phonemes in these sentences using the training set, and then make near perfect detection for these sentences in the test set. This would make the model appear to perform better than it does in reality. After the exclusion of SA sentences, there were 8 sentences per speaker left.

To keep the training and validation set balanced, we ensured a (approximately) 50/50 fricative-to-non-fricative sample ratio. To do this, we extracted 16 randomly placed 192 ms (3072 in samples) long speech segments from each sentence: 8 segments labeled as fricative phoneme and 8 segments labeled as non-fricative phoneme. Recall that the label for a segment is determined based on the identity of the last sample of the segment. If a sentence did not have a fricative phoneme sample, we generated 16 non-fricative segments; this procedure did not affect the (approximate) 50/50 fricative-to-non-fricative sample ratio, since there were not many sentences in training and validation set that had no fricative phoneme sample in them.

For each epoch, our training set consisted of 59136 (462 speakers x 8 sentences x 16 random segments) speech segments placed randomly as explained above. One such segment is highlighted in green in Figure 1. From epoch to epoch the training set of 59136 segments was randomly regenerated from the training set of the TIMIT dataset.

We applied the same procedure to form our validation set, but we kept it fixed for different training runs and also for different epochs in same training run. Our validation samples consisted of 6400 (50 speakers x 8 sentences x 16 segments) speech segments.

For each segment $x = [x_1, \ldots, x_{3072}]$ from training, validation and test sets, we applied standard deviation normalization $x \leftarrow x/\sqrt{\mathrm{Var}(x)}$, where $\mathrm{Var}(x)$ is the variance of the segment.

## 2.2 Network architecture

Table 2 shows the layer structure of the model we used. The input layer of the network takes raw speech segments of 3072 samples (192 ms at 16 kHz sampling rate). The input layer is followed by 5 stages. Stage 1 consists of one convolutional layer, stages 2–5 consist of 6 convolutional layers each. For each convolutional layer, in Table 2, the first number in the square brackets indicates the kernel size of the layer, the number after "/" indicates the stride value in the convolution, and the last number indicates the number of filters used in the layer. The strides are only applied to the first convolutional layer of each stage. All of the layers have *ReLU* non-linearity, except the single neuron at the output layer that has the *sigmoid* non-linearity. In stages 2–5 we used residual connections (He et al., 2016). If the filter dimensions do not match between two layers for residual connection, we applied zero padding to match the filter dimensions (some residual connections had to be applied among two layers that had different number of filters, since we increased the number of filters as we went deeper in the model). The residual connections were applied around pairs of convolutional layers, connecting the input to the first layer of the pair to the output of the linear filter of the second layer of the pair, immediately before the non-linearity of the second layer. After the convolutional stages, 1D global average pooling layer was used. The total number of parameters of this model is 1.1 million. See Figure A.1 in Appendix for details of the model.

Table 2: Network architecture

| LAYER | DESCRIPTION |
|---|---|
| Input | Raw speech segment with shape $[3072, 1]$ |
| Convolutional Stage 1 | [32/6, 48] |
| Convolutional Stage 2 | [8/3, 64] x 6 |
| Convolutional Stage 3 | [8/3, 64] x 6 |
| Convolutional Stage 4 | [8/2, 80] x 6 |
| Convolutional Stage 5 | [8/2, 96] x 6 |
| Global Average | 1D global average pooling |
| Output | Single sigmoid neuron |

## 2.3 Alternative architectures

We experimented widely with the network architecture: the input segment size, the number of layers, the number of features. The network in Table 2 achieved the best performance in our experiments. As can be seen in Figure 2 below, the input size of 3072 samples corresponds to 2-3 phonemes. Intuitively, this provides abundant information to infer the phoneme that follows. Performance of more shallow networks and networks with fewer features is discussed in Section 4.

Our network receives the raw speech signal as input. The use of raw waveform (see also Palaz et al. (2013); Zeghidour et al. (2017)) enables us to learn problem-specific low-level features. Nevertheless, in audio processing it is more common to first transform the raw signal into a time-frequency representation, for example, by computing the Mel-Frequency Cepstrum Coefficients (MFCC), and then to feed the resulting features into a classifier. Because of the time-frequency uncertainty principle, the computation of MFCC features requires a window of certain size, often 25 ms. To compensate for the delay incurred by processing the signal in the window, the classifier must learn to extrapolate detections into the future. While this is possible, as explained in Section 5 below, in general it is a harder task that to make predictions about the present sample. In a separate line of work, we used a proprietary high-quality time-frequency filterbank, followed by a recurrent neural network, to solve the fricative detection problem. After extensive experimentation, we were able to get within a few percent of the performance reported in this paper, however, we were not able to surpass the performance of the model in Table 2. The advantage of the time-frequency transformation followed by a recurrent neural network, is that that approach leads to a much more computationally efficient implementation. The value of this work is to push the boundary of what is possible in terms of detection accuracy. We plan to report the somewhat inferior but computationally more efficient results in a separate publication.

## 2.4 Training

We used *binary crossentropy* loss and *Adam* optimizer starting with 0.001 learning rate. We halved the learning rate if the validation loss did not decrease within 10 epochs, and we applied early stop-

ping if the loss computed on the validation set did not decrease within 40 epochs. We applied 0.0001 l2 weight decay to all of the weights in the convolutional layers, and applied batch normalization after each linear filtering operation and before each non-linearity. See Figure A.1 in Appendix for details on batch normalization. For the results reported in Tables 3, 4 and 5 and Figures A.3 and A.4 we trained the network 10 times and selected the network that performed best on the validation set; for the results reported in Table 7 we trained the network once only.

## 3 EVALUATION

We evaluated the performance of our detection algorithm with the test set (the core test of the TIMIT dataset) that contains 192 sentences with 24 different speakers (each speaker has 8 sentences since we excluded two SA sentences as explained in Section 2.1). The total duration of the test set is 545 seconds.

Figure 2 explains how we use the trained network to make inferences. The raw speech is displayed in blue in the top display. We take a sliding window that consists of 3072 samples (192 ms), represented in black in the top display. The window is moved from left to right, one sample at a time. For each location of the sliding window, we apply the network to the speech segment marked by the window. This gives us the posterior probability for the event that the sample at the end of the sliding window belongs to a fricative phoneme. The posterior probabilities are displayed in blue in the bottom display.

When we compare the network's detections with the ground truth displayed in orange, we observe that the network was able to detect the beginning and the end of the fricative phoneme regions accurately, even though it did not have access to the future context.

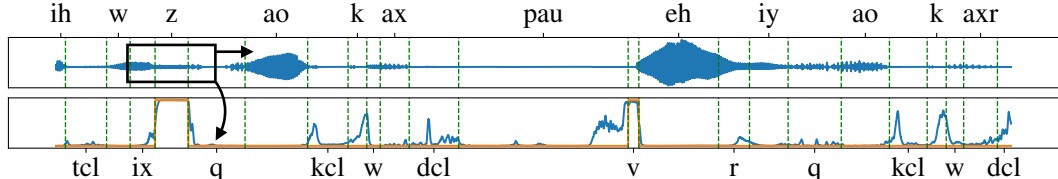

Figure 2: Top: Raw speech in blue and input to the network in the black box. Bottom: Posterior probabilities calculated by the network in blue and the ground truth in orange.

To make the final decision based on the posterior probabilities produced by the network, we apply thresholding as follows:

$$Decision = \begin{cases} \text{Fricative,} & \text{if } p(\boldsymbol{x}) > threshold \\ \text{Non-fricative,} & \text{otherwise.} \end{cases} \tag{1}$$

Above, $p(\boldsymbol{x})$ (the blue line in the lower display in Figure 2) is the posterior probability for the event that the last sample of $\boldsymbol{x}$ (segment in black box in the upper display in Figure 2) belongs to a fricative phoneme. We used the validation set to decide a threshold that maximized the Unweighted Average Recall (UAR) of our network. UAR is the average recall of the two classes (fricative and non-fricative phonemes):

$$UAR = \frac{Recall_f + Recall_n}{2}, \qquad Recall_f = \frac{TP}{TP + FN}, \qquad Recall_n = \frac{TN}{TN + FP}.$$

Above, $TP$, $TN$, $FP$, and $FN$ is the number of true positive, true negative, false positive, and false negative detections of fricative phonemes, respectively; $Recall_f$ and $Recall_n$ is the recall rate for fricative and non-fricative phonemes, respectively. We used UAR as our metric since the test set was highly unbalanced (see *# of Samples* column in Table 3).

We also report the precision rates and the $F_1$-scores ($i \in \{f, n\}$):

$$Precision_f = \frac{TP}{TP + FP}, \quad Precision_n = \frac{TN}{TN + FN}, \quad F_{1,i} = 2 \times \frac{Precision_i \times Recall_i}{Precision_i + Recall_i}.$$

Table 3 summarizes the detection results with the selected threshold (0.42). Overall 94.76% UAR was achieved. To the best of our knowledge, this is the best UAR for fricative phoneme detection on the TIMIT dataset.

Table 3: Precision, Recall and F1-Score in % for our phoneme detection algorithm. Number of Samples indicates the total number of time samples (at 16 kHz) in the test set.

| CLASS | PRECISION | RECALL | F1-SCORE | # OF SAMPLES |
|---|---|---|---|---|
| Non-fricative Phonemes ($n$) | 98.98 | 94.71 | 96.80 | 7344071 |
| Fricative Phonemes ($f$) | 77.15 | 94.80 | 85.07 | 1382200 |
| Unweighted Averages | 88.06 | **94.76** | 90.93 | |

## 3.1 COMPARISON WITH THE LITERATURE

In this section we compared our algorithm with other results in phoneme detection literature. We first summarized the literature that focused only on fricatives phoneme detection (as in this work), then we briefly summarized the literature that focused on generic phoneme identification. Finally, we summarized all the results in one table.

### 3.1.1 FRICATIVE PHONEME DETECTION

Fricative phoneme detection is the task of detecting all occurrences of fricative phonemes in continuous speech. Vydana & Vuppala (2016) investigated the fricative phoneme detection using a combination of *S-Transform* and *Linear Prediction Coefficients (LPC)* approach. In their evaluation, the authors used a post-processing rule to remove spurious detections: if the duration of the detected evidence was shorter than 10 ms, then this evidence was not considered as fricative phoneme. The authors applied majority voting to each phoneme, so that a single detection is made only at the end of each phoneme. Further, the authors evaluated their algorithm by using 40 sentences from the TIMIT dataset, without excluding the SA sentences (see Section 2.1). This, we believe, might have introduced a bias in the reported detection rate making it too optimistic, since the SA sentences are same for all speakers in the dataset and must have appeared both in the training set and in the test set. The authors reported 83.81% recall rate for the fricative phoneme class and did not report the recall rate for the non-fricative class.

Ruinskiy & Lavner (2014) investigated the fricative phoneme detection by combining time domain and frequency domain features that were fed into a linear classifier. The detections were post-processed by using median filtering to avoid short gaps within a sequence of positive detections, and to eliminate detections that were too short to be considered as whole phonemes. The authors applied majority voting for each phoneme and only evaluated their algorithm with *unvoiced* fricative phonemes (/s/, /sh/, /f/, /th/) and not with all 8 fricative phonemes from Table 1. In the evaluation stage, the authors did not exclude the SA sentences, which, as stated above, might have introduced a bias in the reported detection rate making it too optimistic. To the best of our knowledge, this reached the best fricative phoneme detection rate reported on the TIMIT dataset to date: 94.08% UAR.

The non-causal post-processing that was used in Vydana & Vuppala (2016) and Ruinskiy & Lavner (2014) might be suitable for some applications, such as for speech-to-text conversion. However it is not suitable for signal processing in hearing aids, because, as explained in Section 1 these devices must work in real time with minimal delay.

### 3.1.2 GENERIC PHONEME IDENTIFICATION

The generic phoneme identification task is similar to the fricative phoneme detection task, with the only difference that the aim is to identify all phonemes individually. This task is more challenging than the fricative phoneme detection since the number of classes is higher (total-number-of-phonemes-many classes vs. two classes).

It was widely believed that the processing of the input context forward and backward in time from the speech frame to be identified is crucial for success in phoneme identification (Graves & Schmidhuber, 2005). Hence, almost all of phoneme identification studies relied on a system where the speech frame to be identified was located in the middle of a larger input context used by the algorithm to make the decision. This comes with a delay in the phoneme identification. For example, if the input to the algorithm consists of $N$ successive overlapping short speech frames (typically 25 ms long each with overlap of 10 ms) and the detection about the middle frame is made, a delay of $(N - 1)/2$ is accumulated, assuming $N$ is odd. This is illustrated in Figure A.5.

Siniscalchi et al. (2011) and Yu et al. (2012) utilized a single hidden layer Multi Layer Perceptron (MLP) and a Deep Neural Network (DNN), respectively, where the input context had 11 frames and the frame in the middle was identified. Therefore, there were 5 frames delay in the identification. Unfortunately, the authors did not report the frame length in seconds. The authors achieved 93.17% and 96.2% recall rates for fricative phonemes on the Nov92 dataset. Siniscalchi et al. (2013) utilized a DNN, using 310 ms (total time span from *Frame 1* to *Frame N* in Figure A.5) input context that was centered around the frame being detected leading to a 155 ms delay. The authors reported 95.4% recall rate for fricative phonemes on the Nov92 dataset. Chen et al. (2014) employed a DNN, where the input context consisted of 9 frames that was centered around the frame being detected. The authors reported 88.2% recall rate for fricative phonemes on the Switchboard part of the NIST 2000 Hub5 dataset. All these methods introduce significant detection delays and the reported rates are not directly comparable to ours because the rates are not provided for the TIMIT dataset. For convenience of the reader we assembled a summary of these results in Table B.1 in the appendix. Zhang et al. (2019); Chorowski et al. (2015); Tóth (2015) addressed the task of generic phoneme detection on TIMIT using neural networks, unfortunately a comparison with this work is not possible since no fricative-specific results are reported.

Table 4: Fricative phoneme detection rates. $Recall_f$, $Recall_n$ and $UAR$ are given in %.

| RELATED WORK | RECALL$_F$ | RECALL$_N$ | UAR | DELAY |
|---|---|---|---|---|
| Ruinskiy & Lavner (2014) | 89.86 | **98.29** | 94.08 | 80 ms[†] |
| Vydana & Vuppala (2016) | 83.81 | - | - | 80 ms[†] |
| *Our Result* | **94.80** | 94.71 | **94.76** | 0 |

[†] Majority voting post-processing leads to a phoneme-length detection delay. Average length of phoneme in the TIMIT dataset is 80 ms. If there would be no majority voting post-processing, the rates of these two contributions would have been worse than reported in the table.

Vydana & Vuppala (2016) and Ruinskiy & Lavner (2014) aimed at fricative phoneme detection and evaluated their algorithm on the TIMIT dataset as in this work. Table 4 compares the reported results in these two approaches to ours. Ruinskiy & Lavner (2014) is a paper that is most closely related to ours, which calls for a detailed comparison provided next.

The authors of Ruinskiy & Lavner (2014) only attempted to detect *unvoiced fricative phonemes*: /s/, /sh/, /f/ and /th/ and applied majority voting in their evaluation. To make a fair comparison, we also tested our detection with *unvoiced fricative phonemes* only and also applied majority voting. Table 5 summarizes the results. A further comparison in terms of Receiver Operating Characteristic (ROC) curves between our results and those of Ruinskiy & Lavner (2014) is reported in Figure A.3 in Appendix.

Table 5: Comparison between this work and Ruinskiy & Lavner (2014). First row: the rates of Ruinskiy & Lavner (2014) for unvoiced fricative phonemes with majority voting. Second row: the rates for our basic approach that does not rely on any post-processing introducing delay and also is evaluated with all 8 fricative phonemes. Third row: the rates for our basic approach evaluated only with unvoiced fricative phonemes as in Ruinskiy & Lavner (2014). Forth row: the rates for our basic approach followed by majority voting (introducing delay as in Ruinskiy & Lavner (2014)) evaluated with unvoiced fricative phonemes. All the rates are given in %. Our rates reported in Forth row were obtained using the same evaluation strategy as in Ruinskiy & Lavner (2014) except the median filtering (since the details for this step were not given in their paper).

| RELATED WORK | RECALL$_F$ | RECALL$_N$ | UAR |
|---|---|---|---|
| Ruinskiy & Lavner (2014) | 89.86 | **98.29** | 94.08 |
| *Our result:* no post-processing, 8 fricatives | 94.80 | 94.71 | 94.76 |
| *Our result:* no post-processing, unvoiced fricatives | 95.92 | 94.71 | 95.32 |
| *Our result:* majority voting, unvoiced fricatives | **96.94** | 95.87 | **96.41** |

Based on the table, we observed the following.

### 3.1.3 DETECTION OF /V/ AND /DH/

Our detection performance increased when fricatives /v/ and /dh/ were excluded from consideration as was done in Ruinskiy & Lavner (2014), since these two phonemes are *voiced fricative phonemes*. The reason for this is well-known: the fricative phonemes /v/ and /dh/ have a vowel-like structure, and can be easily confused with vowels, reducing $Recall_f$. Figure 3 shows the resemblance of the fricative phonemes /v/ and /dh/ with the vowel phonemes /iy/ in *beet* and /ux/ in *toot* in time-frequency plain.

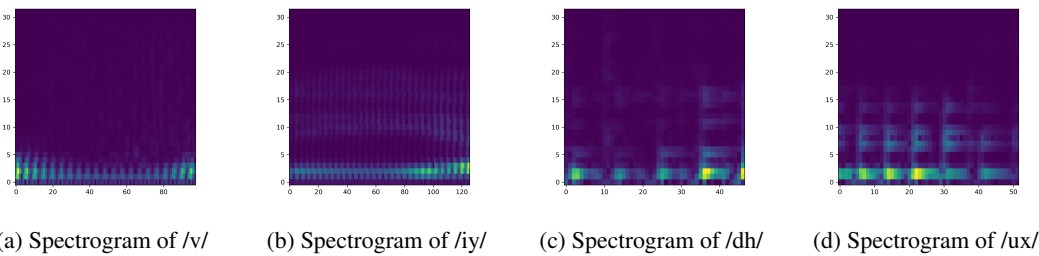

| (a) Spectrogram of /v/ | (b) Spectrogram of /iy/ | (c) Spectrogram of /dh/ | (d) Spectrogram of /ux/ |

Figure 3: Phonemes in time-frequency plain: /v/ resembles /iy/ and /dh/ resembles /ux/.

Figure 4 demonstrates the effect of this resemblance in the detections of our network: In Figure 4a, the posterior probability for fricativity of phoneme /dh/ is not as high as that for fricativity of phoneme /f/. Same problem exists for /v/ as compared to /z/ in Figure 4b.

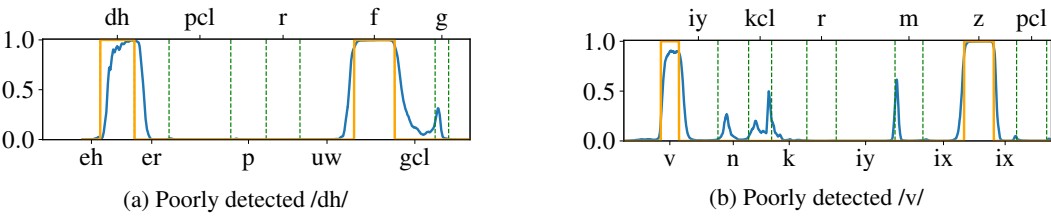

(a) Poorly detected /dh/                  (b) Poorly detected /v/

Figure 4: Posterior probability produced by our network for phonemes /dh/, /f/, /v/ and /z/.

### 3.1.4 MAJORITY VOTING

Majority voting in the evaluation makes a single decision for a phoneme based on all detections for the individual samples constituting that phoneme: if more than half of the samples in the phoneme are classified as fricative by the network, then the phoneme itself is classified as fricative.

Majority-voting-based post-processing introduces a delay equal to the duration of a phoneme making it useless for applications in hearing aids. We applied this post-processing to the detections of our network to make a direct and fair comparison with Ruinskiy & Lavner (2014). As expected, with the post-processing the performance increased to $96.41\%$ UAR rate. This is about 39% decrease in error rate as compared to $94.08\%$ UAR of Ruinskiy & Lavner (2014), a substantial improvement. We stress that even without any post-processing and evaluated with all 8 fricative phonemes our result ($94.76\%$ UAR rate) is still better than those of Ruinskiy & Lavner (2014) that does use post-processing and evaluate only with unvoiced fricative phonemes, albeit by not as much.

## 4 COMPUTATIONAL CONSIDERATIONS

In this paper when we claim that our method introduces zero delay, we make a statistical statement: Detection for the current sample is based only on the past signal. So far, we ignored the computational aspect of the problem: Making inference using a neural network takes time for computation. This introduces a certain delay that depends primarily on how powerful the computer is, how large the neural network is, and how optimized the software is.

On the one hand, the processor in a hearing aid is much weaker than that in a modern powerful GPU. On the other hand, one can envision installing a dedicated low-power chip for neural network-based

inference into future hearing aids; modern phones already contain such chips. No matter what the computational infrastructure is, a smaller model is always more efficient. The basic model described in Table 2 consists of 25 convolutional layers; it is called Net25. We constructed two smaller models as follows. The model Net25h is obtained from Net25 by reducing the number of features in each layer by a factor of two (h stands for half). The model Net19 is obtained from Net25 by removing the Convolutional Stage 5; details of these networks are given in Figure A.2 in the Appendix for reference. Table 6 summarizes the detection rates of the three models, the number of parameters in each model, and the processing time per sample for the Keras implementation executed on Nvidia Quadro P1000 GPU. One can see that Net25h has more than three times fewer parameters as Net25, about 35% faster processing speed, and its UAR rate is about 6% worse than that of Net25.

Table 6: Fricative phoneme detection rates and computational time for different model sizes. Net25 is trained 10 times, Net19 and Net25h are trained 4 times. The reported results are for the networks that perform best on the validation set.

| MODEL | RECALL$_F$ | RECALL$_N$ | UAR | # OF PARAMETERS | ms / SAMPLE |
|-------|-----------|-----------|------|-----------------|-------------|
| Net25 | 94.80 | 94.71 | 94.76 | 1.1M | 1.25 |
| Net19 | 92.81 | 95.95 | 94.38 | 0.7M | 1.04 |
| Net25h | 94.82 | 94.03 | 94.42 | 0.3M | 0.82 |

Still, there will always be a nonzero time that is necessary to perform the computation. Since we are targeting zero delay, we must accommodate the extra time needed. A natural solution is to try to extrapolate our detections into the future. This is done in the next section.

## 5 EXTRAPOLATING DETECTIONS INTO THE FUTURE

In this section, we use our network to extrapolate fricative phoneme detections into the future. In Figure 5, the green highlighted segment represents the input to the network. The goal of the network is to use this input to calculate the probability that a sample (marked in orange) located a couple of ms in the future, beyond the right boundary of the green segment (marked in red), belongs to a fricative phoneme.

To accomplish this goal we retrain the network with the data that is relabeled accordingly. For example, the label for the segment highlighted in green in Figure 5 would be based on the identity of the orange sample, not on the identity of the red sample as was done previously.

We considered four different values for the extrapolation gap: detecting 1ms, 2ms, 3ms, and 4 ms ahead into the future. This corresponds to the time difference between the orange line (future) and the red line (present time) in Figure 5.

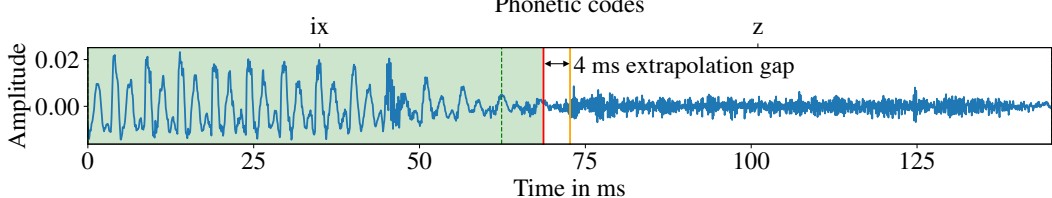

Figure 5: Input segment in green and sample to be detected in orange. Extrapolation gap is 4 ms.

Except for the relabeling of the training set, the training is exactly the same as before. The inference is also unchanged, except that to make a detection for the sample at orange line, we fed to the network the green highlighted segment in Figure 5. Table 7 reports the rates for the fricative phoneme detection as a function of the extrapolation gap.

Table 7 clearly shows that the performance decays very gracefully with the increasing extrapolation gap. With the extrapolation gap of 1-2ms we already have enough computational time for inference. Even at 3ms extrapolation gap and solving a harder problem of detecting all 8 fricative phonemes our method still outperforms the method of Ruinskiy & Lavner (2014) in UAR.

Table 7: Rates for fricative phoneme detection as a function of extrapolation gap reported in %.

| Extrapolation gap | $RECALL_F$ | $RECALL_N$ | UAR |
|---|---|---|---|
| 0 ms | 94.80 | 94.71 | 94.76 |
| 1 ms | 95.04 | 93.68 | 94.36 |
| 2 ms | 93.25 | 95.47 | 94.36 |
| 3 ms | 94.33 | 94.21 | 94.27 |
| 4 ms | 95.01 | 92.76 | 93.89 |

## 6 CONCLUSION

In this work, we provided a state-of-the-art baseline for fricative phoneme detection and demonstrated that this problem may be solved with zero delay. Further, we demonstrated that it is feasible to extrapolate fricative phoneme detections into the future, creating sufficient computational time for inference. Among other things, these results are important for building the future generation of hearing aids. Our results are fully reproducible and we make the complete source code available. We also include all the trained models that were used in evaluations in this paper.

The following interesting research directions remain. Restructure the computational graph of the evaluation pipeline to take advantage of the time-series-like structure of the data and avoid redundant computations. Reduce the size of the network to make it more suitable for a low-power device (hearing aids). This may be achieved, for example, using deep learning compression techniques: Pruning, compression through quantization, knowledge transfer through distillation or teacher-student approach, using memory efficient models such as fire module or depth wise convolutions.

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

# A  SUPPLEMENTARY DETAILS

Table A.1: Phonemes in Figure 1.

| PHONETIC CODE | /ih/ | /tcl/ | /w/ | /ix/ | /z/ |
|---|---|---|---|---|---|
| EXAMPLE WORD | bit | it | way | debit | zone |

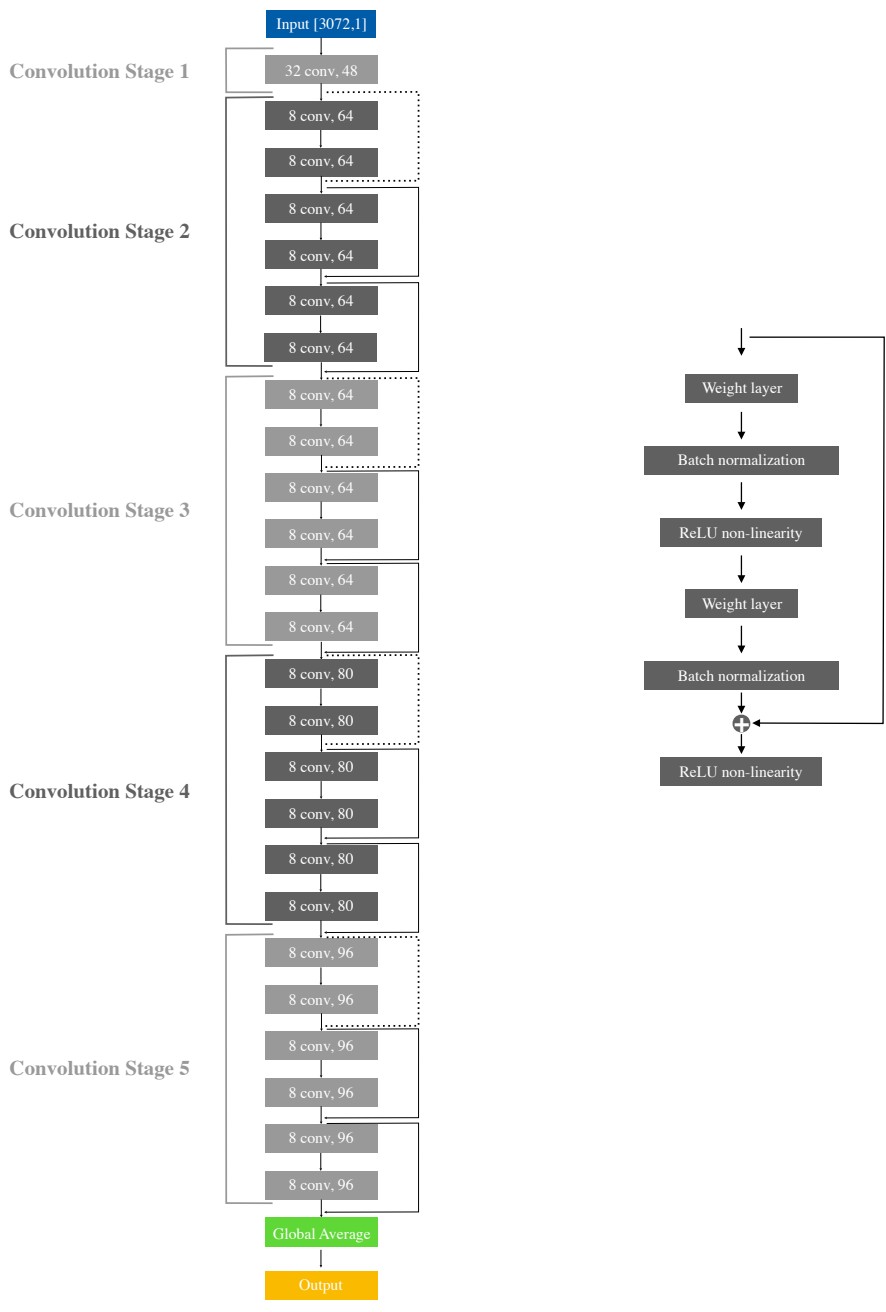

Figure A.1: Left: Detailed network architecture used in the main paper consists of 25 convolutional layers; it is called Net25. The dashed lines show the downsampling for residual connections and Global Average represents 1D Global Averaging. Right: Illustration of residual connection; weight layer represents the convolutional layer without non-linearity.

Apart from the main network displayed in Figure A.1, we considered smaller model sizes displayed in Figure A.2.

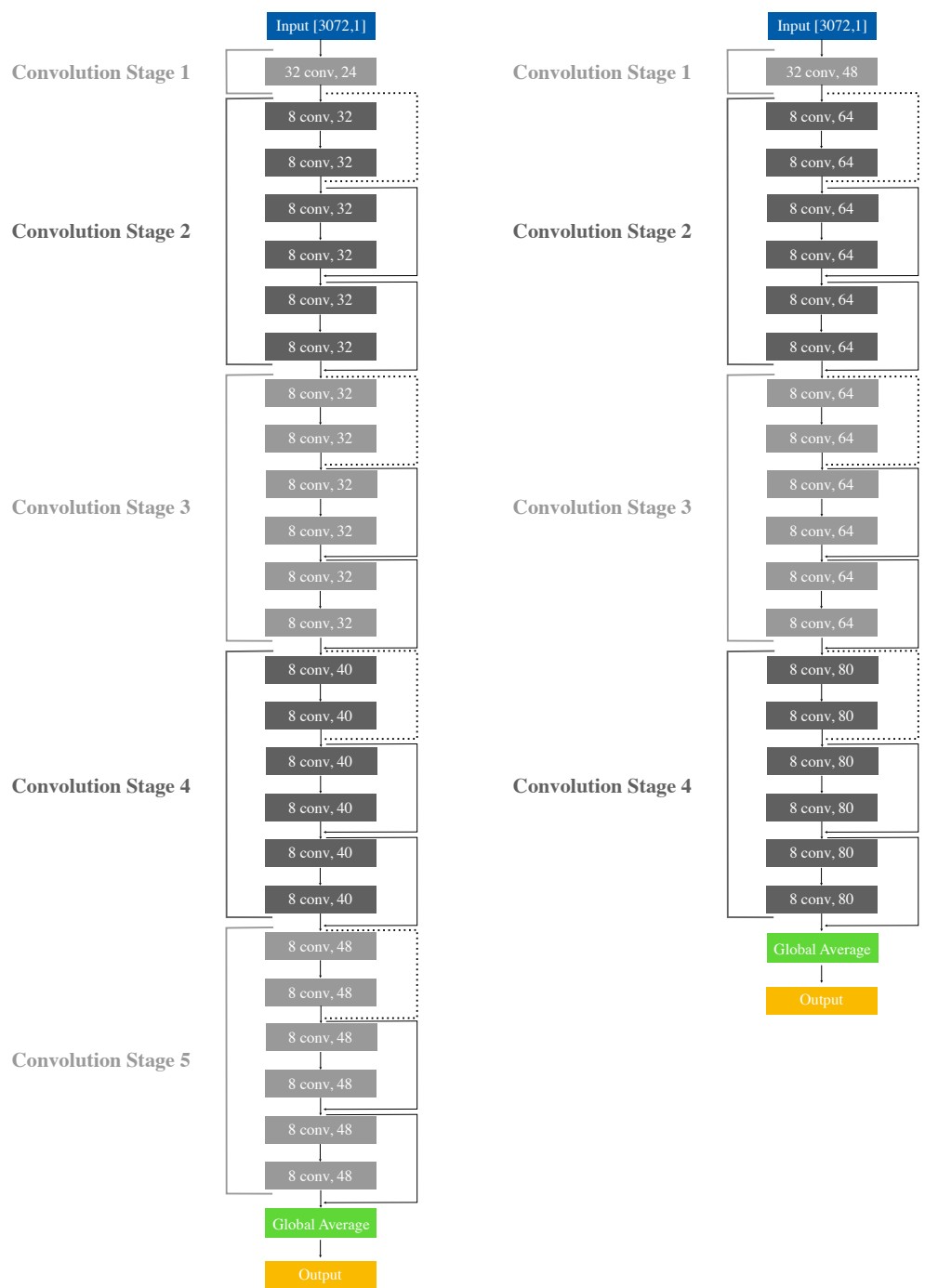

Figure A.2: Left: Network architecture with half the number of filters in each convolutional layer as in Net25; it is called Net25h. Right: Network architecture with the same number of filters as in Net25 obtained by removing the Convolutional Stage 5 from Net25 consists of 19 convolutional layers; it is called Net19.

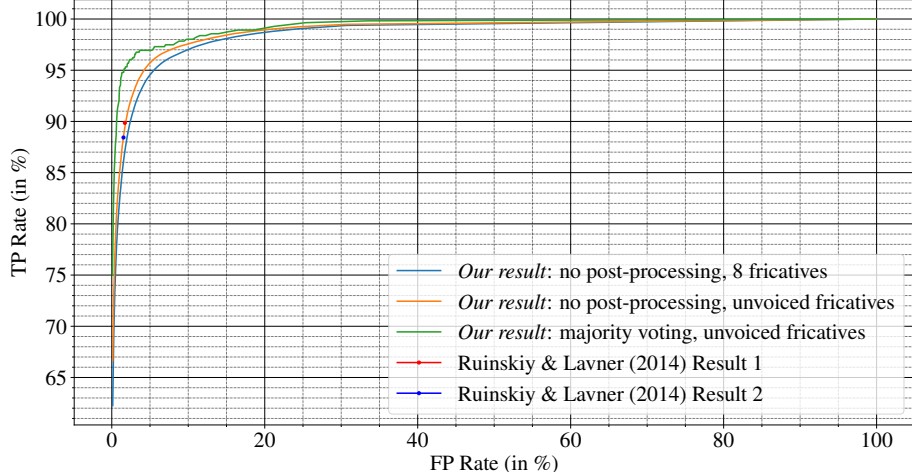

Figure A.3: Comparison of our results with Ruinskiy & Lavner (2014) using ROC curves. To obtain the ROC curve, we applied the decision rule in Equation 1 with thresholds in the range $[0, 1)$ with increment of $0.01$. Note that there are two rates given in the figure for Ruinskiy & Lavner (2014), since they reported two different evaluations. In the main part of the paper we only used Result 1 of Ruinskiy & Lavner (2014) (in red) in comparisons since it has better UAR than the Result 2 (in dark blue). The zoomed version of ROC curves is on next page.

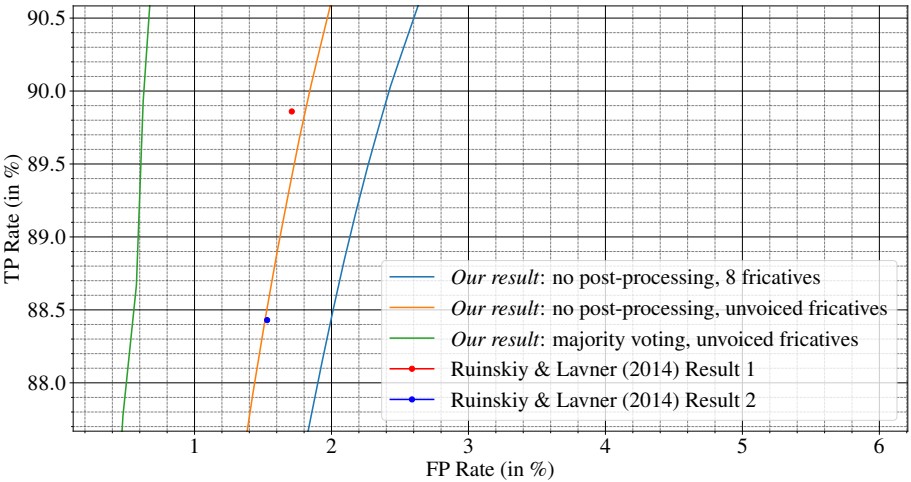

Figure A.4: Zoomed version of Figure A.3.

Table A.2: $Recall_f$ in % on fricative phoneme detection task for related approaches. In all these contributions the primary task is generic phoneme identification, but the results are evaluated on the fricative phoneme detection task. Delays are given in frames if they were not stated in the original work in seconds. Note that these contributions are *not* evaluated on the TIMIT dataset.

| RELATED WORK | RECALL$_F$ | DELAY | DATASET |
|---|---|---|---|
| Siniscalchi et al. (2011) | 93.17 | 5 frames | Nov92 |
| Yu et al. (2012) | 96.2 | 5 frames | Nov92 |
| Siniscalchi et al. (2013) | 95.4 | 155 ms | Nov92 |
| Chen et al. (2014) | 88.2 | 4 frames | NIST 2000 Hub5 |

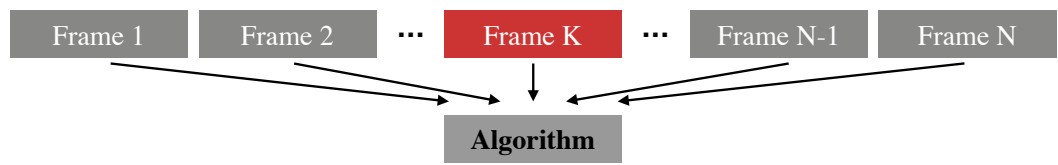

Figure A.5: Detecting phonemes in the middle of a larger context. Input context $\{Frame\ 1, Frame\ 2, \ldots, Frame\ N\}$ to the detection algorithm consists of overlapping successive speech segments. The speech segment to be identified is in the middle of the input context.

## B  DETECTING PHONEME BOUNDARIES

Our networks were optimized to solve the two class classification problem on the level of samples: does the current sample belong to a fricative phoneme or not? A related problem is to precisely identify the boundaries of fricative phonemes. A good metric to measure the quality of boundary identification has been proposed in Räsänen et al. (2009). The metric is called $R$-value and it is defined as follows:

$$R_{val} = 1 - \frac{\sqrt{(1-R)^2 + OS^2} + \left| \frac{Recall_{fb} - 1 - OS}{\sqrt{2}} \right|}{2}, \quad OS = \frac{Recall_{fb}}{Precision_{fb}} - 1. \quad (2)$$

where $Recall_{fb}$ and $Precision_{fb}$ denote the recall and the precision in identifying the fricative boundary correctly. The boundary is identified correctly if it lies no further than $threshold_b$ milliseconds from a fricative boundary in the ground truth data. Each ground truth boundary may only match a single boundary in the data.

Interesting work on identifying phoneme boundaries precisely (not specializing to fricative phonemes) has been done in Franke et al. (2016); Michel et al. (2017); Adi et al. (2016). In the table below we report the performance of Net25 on this task for fricative phonemes. We note that

Table B.1: Rates of Net25 on detection of fricative boundaries task. The boundaries for the beginning and the end of a phoneme are detected and compared to the ground truth separately. We report the average scores over beginning of phoneme/end of phoneme boundaries.

| threshold$_B$, ms | PRECISION$_{FB}$ | RECALL$_{FB}$ | F1$_{FB}$ | R-VAL$_{FB}$ |
|---|---|---|---|---|
| 20 | 0.58 | 0.77 | 0.66 | 0.60 |
| 25 | 0.61 | 0.81 | 0.70 | 0.63 |
| 30 | 0.63 | 0.83 | 0.72 | 0.64 |

our network was not trained for this task and the results do not seem to be state-of-the-art. We leave it to future work to identify and remove the dominant sources of errors in this task and to understand how critical the precise boundary detection is for the applications in hearing aid devices.

