# OpenReview forum: "FRICATIVE PHONEME DETECTION WITH ZERO DELAY"
_ICLR.cc/2020/Conference — Reject_

### Official Review · AnonReviewer2 · 2019-10-22
**Official Blind Review #2**

**Rating:** 3

**Review:**


## Updated review

I have read the rebuttal. I have some concerns about the new version of the paper: The addition of section 2.3 about the MFCCs is welcome but feels a bit out of place. The first part about the MFCC is interesting and relevant, but it could be in the introduction as a motivation. The second part about the "proprietary high-quality time-frequency filterbank" is not clear at all. Firstly, results are discussed so this part should be in the Evaluation section. Secondly, why using proprietary filterbanks and not the standard Mel filterbanks ?

Given that the rebuttal and the new version of the paper didn't address my major concerns, I am keeping my original rating.

## Original review

This paper presents an approach to detect fricative phoneme in speech with as little delay as possible, in the context of hearing aids improvement. The model is based on CNN and is trained to detect fricative given the past context. The model is evaluated in terms of recall and compared with recent published works. The results show that the proposed approach outperforms the baselines and yields state-of-the-art performance with no delay. The paper concludes with some analysis on the computational cost and draw possible future work.

This paper should be rejected for the following reasons:
- The novelty is very limited: this work applied a well-known architecture (CNN) to a common problem, phoneme recognition. This only novelty is the zero-delay constraint, which is probably not sufficient for ICLR.
- The significance is also limited given the very specialized application.
- Some references are missing (see below).
- The presented results are not very clear.
- The computational considerations section is interesting but is missing some important elements.

Detailed comments:
- The authors selected raw speech signal as input to the CNN, which is not trivial and should be motivated and discussed in the paper. For instance, using the standard features like Mel filterbanks or MFCC will introduce a delay as they are computed on an overlapping window of 25ms. Phoneme recognition using raw speech as input to a CNN has been presented before, the authors should cite [1] and [2] for instance.
- Table 4 is confusing as the first four lines are not actually evaluated on TIMIT, so I don't see the point of adding these numbers to the table, as they cannot be compared anyway. I would remove these four lines from the Table.
- In terms of previous works, phoneme recognition on TIMIT is a very popular task, and many others could be cited, such as [3-5].
- On the computation consideration, the analysis is interesting, but a discussion on the size (i.e. number of parameter) of the network is missing: one way to decrease computation time is to have a smaller network, which is in line with the application: hearing aids probably do not have gigabytes of ram available.
- Question about the network: the input segment seems to be of size 3072 samples, why ? any motivation for this particular input size ?

My review can seem to be a bit harsh, I actually enjoyed the paper, but I don't think ICLR is the right conference for it, and I would advise the authors to improve it and submit it to a speech conference.

References:
[1] Palaz, D., Magimai Doss, M. and Collobert, R.. "Estimating Phoneme Class Conditional Probabilities from Raw Speech Signal using Convolutional Neural Networks." Proceedings of Interspeech 2013.
[2] Zeghidour, N., Usunier, N., Kokkinos, I., Schaiz, T., Synnaeve, G., & Dupoux, E. "Learning filterbanks from raw speech for phone recognition". Proceedings of ICASSP 2018.
[3] Zhang, Ying, Mohammad Pezeshki, Philémon Brakel, Saizheng Zhang, Cesar Laurent, Yoshua Bengio, and Aaron Courville. "Towards end-to-end speech recognition with deep convolutional neural networks." arXiv preprint arXiv:1701.02720 (2017).
[4] Chorowski, Jan K., et al. "Attention-based models for speech recognition." Advances in neural information processing systems. 2015.
[5] Tóth, László. "Phone recognition with hierarchical convolutional deep maxout networks." EURASIP Journal on Audio, Speech, and Music Processing 2015.1 (2015): 25.

**Experience Assessment:**

I have published in this field for several years.

**Review Assessment: Checking Correctness Of Derivations And Theory:**

N/A

**Review Assessment: Checking Correctness Of Experiments:**

I carefully checked the experiments.

**Review Assessment: Thoroughness In Paper Reading:**

I read the paper at least twice and used my best judgement in assessing the paper.

---

> ### Author Response · Authors · 2019-11-15
> **Thank you and review reply**
>
> We would like to thank the reviewer for carefully evaluating our work and for providing very thoughtful comments. Surely, these comments have already helped us improve the paper. We are also happy that the reviewer enjoyed the paper.
>
> Below we give answers to the specific comments of the reviewer.
> Q1: The authors selected raw speech signal as input to the CNN, which is not trivial and should be motivated and discussed in the paper. For instance, using the standard features like Mel filterbanks or MFCC will introduce a delay as they are computed on an overlapping window of 25ms. Phoneme recognition using raw speech as input to a CNN has been presented before, the authors should cite [1] and [2] for instance.
>
> A1: We added Section 2.3 to the paper discussing this issue. Shortly, we did experiment with MFCC-type approach followed by a recurrent neural network. We were not able to achieve the quality of Net25 on the task, but we were able to get within 5% of the accuracy of Net25 with a propitiatory filterbank and much more computationally efficient processing. We plan to publish these results in another paper.  We cited [1] and [2] as the reviewer suggested.
>
> Q2: Table 4 is confusing as the first four lines are not actually evaluated on TIMIT, so I don't see the point of adding these numbers to the table, as they cannot be compared anyway. I would remove these four lines from the Table.
> A2: We agree. We moved this part of the table to the appendix, just for reference of an interested reader.
>
> Q3: In terms of previous works, phoneme recognition on TIMIT is a very popular task, and many others could be cited, such as [3-5].
> A3: We cited [3-5] appropriately as the reviewer suggested.
>
> Q4: On the computation consideration, the analysis is interesting, but a discussion on the size (i.e. number of parameter) of the network is missing: one way to decrease computation time is to have a smaller network, which is in line with the application: hearing aids probably do not have gigabytes of ram available.
> A4: We added a table with all the relevant details into Section 4 of the paper. It was already stated in the paper that Net25 has 1.1M parameters. Also, we have experimented with a much smaller recurrent neural network, please see answer to Q1.
>
> Q5: Question about the network: the input segment seems to be of size 3072 samples, why ? any motivation for this particular input size?
> A5: We discussed this more in the paper: the choice is empirical, based on experimentation. The input window covers 2-3 preceding phonemes.

---

### Official Review · AnonReviewer3 · 2019-10-22
**Official Blind Review #3**

**Rating:** 6

**Review:**

This paper apples supervised deep learning methods to detect exact duration of a fricative phoneme in order to improve practical frequency lowering algorithm. A major challenge compared to existing work is to have an algorithm with nearly zero delay while preserving detection accuracy. A deep convolutional neural network is trained for this purpose and it is validated on TIMIT dataset.

After a careful preprocessing of the data, long segments of raw audio are given as input to the convolutional net. It is trained as a binary classification problem. Therefore, for each different phenome, a different network is needed. To improve the accuracy, Majority voting is also adopted. This however increases the computational cost. To address this issue, an extrapolation detection problem is formulated to predict the fricative phoneme a few ms in advance. Extensive numerical results show that the approach still outperforms the method of Ruinskiy & Lavner (2014) in Unweighted Average Recall.

I find the accuracy attained by the neural nets quite impressive, although more insights would be favored to understand what is going on. This is yet an interesting application of deep learning useful for real-life problems. If the method could be tested on another dataset, the result would be more convincing.


**Experience Assessment:**

I do not know much about this area.

**Review Assessment: Checking Correctness Of Derivations And Theory:**

N/A

**Review Assessment: Checking Correctness Of Experiments:**

I assessed the sensibility of the experiments.

**Review Assessment: Thoroughness In Paper Reading:**

I read the paper at least twice and used my best judgement in assessing the paper.

---

> ### Author Response · Authors · 2019-11-15
> **Thank you**
>
> We would like to thank the reviewer for the positive evaluation of our work. We agree that evaluating our method on more datasets will make the work more solid. Unfortunately, we were not able to accomplish this quickly enough to add to the paper at this point, but we will do this in the future.

---

### Official Review · AnonReviewer1 · 2019-10-23
**Official Blind Review #1**

**Rating:** 3

**Review:**

====================================== Updated Review =====================================
I would like to thank the authors for providing more experiments and details regarding their work.
However, after reading the authors rebuttal, I still think that there is more work to do in terms of comparison to prior work. That way it would be much clearer what is contribution of this work, and how it can be used for future research in that field.

Hence, I would like to keep my score as is.
==========================================================================================

This paper describes a method for fricative phonemes boundary detection with zero delays. The authors suggest optimizing a convolutional based neural network with a binary cross loss function to detect such events.
The authors provide results on the TIMIT dataset and compare the proposed model to several baselines.

The task of phoneme boundary detection was well studies under different setups and is very important for various applications, including the one proposed in this paper.

However, I have some major concerns regarding this paper, which I would like the authors to clarify. Without these, it is hard to understand the contribution in this paper.

1) the authors chose to model the problem from the raw wave. Although it is getting popularity in several speech processing tasks, it is not clear why not using magnitude/MFCC, for example. In case the authors claim that learning from the waveform is better, I suggest providing a comparison to other features.
Additionally, did the authors experience with simpler architectures Maybe more shallow models? Regarding supervision, did the authors tried comparing to the method proposed by [2] but with a unidirectional RNN? Similar to [3].

2)  If I understand it correctly, the motivation for this task was: accurate detection of fricatives boundary can be used to shift into lower frequency bands in hearing aids. It seems like the boundaries are more important than other phoneme parts such as a mid phoneme, for example.
In that case, a better metric might be Presicion + Recall + F1 + R-val next to the boundaries (for instance, with a tolerance level of 10-20ms). Those metrics were suggested on several studies of phoneme segmentation, [1], [2].

3) The comparison in Table 3 is very strange. Results are reported on different datasets. Although the authors mentioned it in the caption, it is still misleading. I suggest the authors to compare either obtain results on the same benchmark or compare to other baselines.

Minor comments:
"If for the majority of the samples in a phoneme our network’s output is greater than the threshold we set" -> not a clear sentence.

[1] Franke, Joerg, et al. "Phoneme boundary detection using deep bidirectional lstms." Speech Communication; 12. ITG Symposium. VDE, 2016.
[2] Michel, Paul, et al. "Blind phoneme segmentation with temporal prediction errors." arXiv preprint arXiv:1608.00508 (2016).
[3] Adi, Yossi, et al. "Automatic Measurement of Voice Onset Time and Prevoicing Using Recurrent Neural Networks." INTERSPEECH. 2016.

**Experience Assessment:**

I have published in this field for several years.

**Review Assessment: Checking Correctness Of Derivations And Theory:**

N/A

**Review Assessment: Checking Correctness Of Experiments:**

I carefully checked the experiments.

**Review Assessment: Thoroughness In Paper Reading:**

I read the paper thoroughly.

---

> ### Author Response · Authors · 2019-11-15
> **Thank you and review reply**
>
> We would like to thank the reviewer for carefully reading the paper and providing very helpful comments.
> We improved the paper based on these comments and submitted a revision. Below we address the specific comments of the reviewer one-by-one.
>
> Q1: The authors chose to model the problem from the raw wave. Although it is getting popularity in several speech processing tasks, it is not clear why not using magnitude/MFCC, for example. In case the authors claim that learning from the waveform is better, I suggest providing a comparison to other features.
> Additionally, did the authors experience with simpler architectures Maybe more shallow models? Regarding supervision, did the authors tried comparing to the method proposed by [2] but with a unidirectional RNN? Similar to [3].
> A1: We added Section 2.3 to the paper discussing this issue. Shortly, we did experiment with MFCC-type approach followed by a recurrent neural network. We were not able to achieve the quality of Net25 on the task, but we were able to get within 5% of the accuracy of Net25 with a propitiatory filterbank and much more computationally efficient processing. We plan to publish these results in another paper.
> -------
>
> Q2:  If I understand it correctly, the motivation for this task was: accurate detection of fricatives boundary can be used to shift into lower frequency bands in hearing aids. It seems like the boundaries are more important than other phoneme parts such as a mid phoneme, for example.
> In that case, a better metric might be Presicion + Recall + F1 + R-val next to the boundaries (for instance, with a tolerance level of 10-20ms). Those metrics were suggested on several studies of phoneme segmentation, [1], [2].
>
> A2: We added an evaluation using these metrics in appendix B. For 20 ms threshold on each side, the r-value we achieved is 0.6. Note that the network is not optimized for this task. We did not have sufficient time to dig deeper into sources of errors on these metrics, therefore these results are in the appendix and not in the main paper. Also, it looks like the r-val in formula (3) in [2] has a typo in (R+1 -OS)/sqrt(2).  The correct formula from [Räsänen, Laine, Altosaar] has r2 = (􏰎-OS + R - 􏰎1)/sqrt(2).
> -------
>
> Q3: The comparison in Table 3 is very strange. Results are reported on different datasets. Although the authors mentioned it in the caption, it is still misleading. I suggest the authors to compare either obtain results on the same benchmark or compare to other baselines.
> A3: We agree. We moved this part of the table to the appendix, just for reference of an interested reader.
> -------
>
> Q4: Minor comments:
> "If for the majority of the samples in a phoneme our network’s output is greater than the threshold we set" -> not a clear sentence.
> A4: We rephrased this sentence.
>
> [2] Michel, Paul, et al. "Blind phoneme segmentation with temporal prediction errors." arXiv preprint arXiv:1608.00508 (2016).

---

### Decision · Program_Chairs · 2019-12-19

**Decision:**

Reject

**Comment:**

The reviewers appreciate the importance of the problem, and one reviewer particularly appreciated the gains in performance. However, two reviewers raised concerns about limited novelty and missing comparisons to prior work. While the rebuttal helped address these concerns, the novelty is still limited. The authors are encouraged to revise the presentation to clarify the novelty.